# Generalized Wardowski type contractive mappings in *b*-metric spaces and some fixed point results with applications in optimization problem and modeling biological ecosystem

**Maryam Iqbal**[1], **Afshan Batool**[1], **Aftab Hussain**[2]*, **Hamed Al Sulami**[2]

**1** Department of Mathematical Sciences, Fatima Jinnah Women University, Rawalpindi, Islamic Republic of Pakistan, **2** Department of Mathematics, King Abdulaziz University, Jeddah, Saudi Arabia

* aniassuirathka@kau.edu.sa

**Data Availability Statement:** All relevant data for this study are within the paper.

## Abstract

In the realm of *b*-metric spaces, this study introduces a novel generalized Wardowski-type quasi-contraction, denoted as $\beta$-($\theta$, $\vartheta$). We begin by applying this new contraction to derive standard fixed point results. Subsequently, we establish the existence of a generalized quasi-contraction of the Wardowski type, thereby validating the robustness of our findings. Specifically, we utilize Nadler's work to model biological ecosystems and apply our results to solve an optimization problem. To illustrate the practical implications and effectiveness of our approach, we provide a comparative analysis between our results and those of Nadler. This comprehensive study underscores the significance and utility of our generalized contraction in both theoretical and applied contexts.

## 1 Introduction

Fixed point, which is referred to as FP theory plays a crucial role in the advancement of nonlinear functional analysis, offering substantial insights and applications across various fields. Its flexibility and wide-ranging applicability make it a vibrant area of research. FP theory investigates the conditions under which a self-map on a set can possess a FP, providing significant theoretical and practical results. One of the most profound contributions to FP theory was made by the renowned mathematician Stefan Banach. His ground breaking result [1] established that in a complete metric space, which is referred to as CMS contractive mappings have a unique FP. This principle, known as the Banach Contraction Principle, which is referred to as BCP has become a fundamental theorem with extensive applications in mathematics and related disciplines. The BCP has been modified and extended in numerous ways to address various mathematical and real-world problems, (see for example [2]).

The BCP's utility spans many scientific and technical domains, leading to the development of various contractive mappings in different types of MS. Recent advancements, such as those by Hussain et al. [3–6], highlight significant progress in both theoretical and applied aspects of FP theory. A notable extension of this work is the introduction of *b*-MS initially

**Funding:** The author(s) received no specific funding for this work.

**Competing interests:** The authors declare that they have no competing interest.

conceptualized by Bakhtin [7] and Bourbaki [8], and later formalized by Czerwik [9]. A $b$-MS is a generalization of the traditional MS that relaxes the strict triangle inequality. In a $b$-MS, the distance function $d(q_1, q_2)$, satisfies a weaker form of the triangle inequality, which can be expressed as:

$$d(q_1, q_3) \leq b[d(q_1, q_2) + d(q_2, q_3)],$$

where $b \geq 1$ is a constant. This relaxation allows for broader applications and provides new insights into FP theory. In recent developments, Kamran et al. [10] extended the FP results to self-mappings in generalized $b$-MSs, further expanding the applicability of the BCP. Subsequent research has explored FP results in $b$-MSs for both single-valued and multi-valued operators (see [11–20]). These advancements in $b$-MSs highlight a significant shift from traditional MSs, offering new perspectives and tools for solving problems in FP theory. The study of $b$-MSs continues to contribute to the broader understanding of FP and their applications, underscoring the ongoing relevance and dynamism of this field.

In this paper, we extend the FP results of Wardowski [21] to $b$-MSs, which are a generalization of MSs. To achieve this, we introduce the concept of $\beta$-$(\theta, \vartheta)$-contraction, incorporating the coefficient $b$ from $b$-MSs into our analysis. Our results demonstrate a broader applicability compared to Wardowski's findings.

The paper is organized into two main sections:

In the first section, we address gaps in the existing literature by applying Nadler's work to provide concrete examples. We also compare the efficiency of our results with those of Nadler, highlighting the advancements and improvements our approach offers.

The second section focuses on our FP results for $\beta$-$(\theta, \vartheta)$-contractions within complete $b$-MSs. We provide examples and applications, particularly in optimization problems, to demonstrate the effectiveness and advantages of our results.

The final section summarizes our findings and conclusions, emphasizing the contributions and implications of our research.

## 2 Preliminaries

This section covers the preliminary findings and analysis that support the article's main conclusions.

**Definition 2.1**. [10] *Let $\chi$ be a nonempty set, and let d be a function. The function d: $\chi \times \chi \to [0, \infty)$ is called b-metric if it satisfies the following conditions*:

1. $0 \leq d(q_1, q_2)$ *and* $d(q_1, q_2) = 0$ *if and only if* $q_1 = q_2$,

2. $d(q_1, q_2) = d(q_2, q_1)$,

3. $d(q_1, q_3) \leq b[d(q_1, q_2) + d(q_2, q_3)]$ *for some* $b \geq 1$,

   *for all* $q_1, q_2, q_3 \in \chi$. *The pair* $(\chi, d)$ *is called b-MS with coefficient b.*

All MSs can be considered as $b$-MS with $b = 1$. The class of $b$-MSs encompasses a broader range than that of MSs, and the concept of a $b$-MS generalizes the notion of an MS. Specifically, the $b$-MS framework is a generalization that is less restrictive than the traditional MS framework. While conditions (1) and (2) in a $b$-MS are similar to those in an MS condition (3), introduces a crucial component. Mastery of the effective application of condition (3) is essential for fully understanding this concept. For a clearer illustration of the importance of the third criterion, see Example 2.2.

**Example 2.2**. [9] *Let $L_p[0, 1]$ be the space of all real functions. Let $p \in (0, 1)$ and $q(O)$ be such that $\int_0^1 |q_1(O)|^p dO < \infty$.*

*Define d: $L_p[0, 1] \times L_p[0, 1] \to [0, \infty)$ as:*

$$d(q_1, q_2) = (\int_0^1 |q_1(O) - q_2(O)|^p)^{\frac{1}{p}},$$

*for each $\in L_p[0, 1]$ of $q_1, q_2$. Then, $(L_p[0, 1]d)$ is a b-MS and $b = 2^{\frac{1}{p}}$ is b-metric coefficient.*

**Definition 2.3**. [10] *Let $(\chi, d)$ be a b-MS with $b \geq 1$.*

1. *If there exists $\dot{\omega} \in \chi$ such that $d(\dot{\omega}_j, \dot{\omega}) \to 0$ as $j \to \infty$, then the sequence $\dot{\omega}_j \subset \chi$ is said to be b-convergent.*

2. *If $d(\dot{\omega}_j, \dot{\omega}_i) \to 0$ as $j, i \to \infty$, then a sequence $\dot{\omega}_j \subset \chi$ is a b-Cauchy sequence.*

3. *A b-MS $(\chi, d)$ is said be complete b-MS if every b-Cauchy sequence in $\chi$ is b-convergent.*

Hausdorff originally presented the idea of the Hausdorff metric or Hausdorff distance in his work Grundzuge der Mengenlehre [22]. Pompeiu-Hausdorff distance is the second name for the Hausdorff distance. In the realm of computers, there are numerous uses for the Hausdorff distance. In computer vision, the Hausdorff distance is used to locate a specified template in any target image. The Hausdorff metric is most frequently used in computer graphics to assess the difference between two different representations of the same 3D object when creating the level of detail required for the effective display of complex 3D models.

**Definition 2.4**. [15] *Let $(\kappa, d)$ be a MS and let $K(\kappa)$ be the class of all nonempty compact subsets of $\kappa$. $\zeta: K(\kappa) \times K(\kappa) \to [0, \infty)$ is the definition of this mapping:*

$$\zeta(\kappa, Z) = \max\{\sup_{q_1 \in \kappa} d(q_1, Z), \sup_{q_2 \in Z} d(q_2, \kappa)\}, \forall \kappa, Z \in K(\kappa)$$

*is referred to as the Pompeiu-Hausdroff metric, induced by metric d and $d(q_1, Z) = \inf\{d(q_1, q_2): q_2 \in Z\}$ is the distance from $q_1$ to $Z \subseteq \kappa$.*

**Example 2.5**. *Let $\mathbb{R}$ is a set of real numbers with metric $d(q_1, q_2) = \sqrt{|q_1 - q_2|}$ with $b = 1$. Then for any two closed intervals $[Q, q_1]$ and $[q_2, q_3]$, we have*

$$\zeta([Q, q_1], [q_2, q_3]) = \max\{\sqrt{|Q - q_2|}, \sqrt{|q_1 - q_3|}\}.$$

**Definition 2.6**. [21] *Suppose that $\Psi$ is collection of all functions $\vartheta : \mathbb{R}^+ \to \mathbb{R}$ with following conditions:*

1. *The value of $\vartheta$ is strictly increasing.*

2. *In the interval $(0, +\infty)$, for any sequence $\{\sigma_j\}$, the following condition holds:*

$$\lim_{j \to \infty} \sigma_j = 0 \Leftrightarrow \lim_{j \to \infty} \vartheta(\varrho_j) = -\infty.$$

3. *$\lim_{\sigma \to 0^+} \sigma^O \vartheta(\varrho_j) = 0$ exists for $O \in [0, 1]$.*

**Definition 2.7**. [21] *Let $(\kappa, d)$ be a MS. If there exist $p \in \mathbb{R}^+$ and $\vartheta \in \Xi$ such that for any $q_1, q_2 \in \kappa$, we have a mapping $\Phi: \kappa \to \kappa$ that is $\vartheta$-contraction.*

$$d(\Phi_{q_1}, \Phi_{q_2}) > 0 \to p + \vartheta(d(\Phi_{q_1}, \Phi_{q_2})) \leq \vartheta(d(q_1, q_2)).$$

Remember that a contraction is always a $\vartheta$-contraction.

**Theorem 2.8**. [21] *Assume that $\kappa$ is a CMS and we have $\vartheta$-contraction $\Phi\colon \kappa \to \kappa$. Then, for every point $\Im \in \kappa$, the sequence $\{F^j\Im\}$ converges to $\Re$, and $\Phi$ has a single FP $\Re \in \kappa$.*

In 2012, Samet *et al.* [23] defined $\beta$-admissible for single-valued mappings.

**Definition 2.9**. [21] *Assume that the set $\kappa$ is not empty.*

1. When a mapping $\beta\colon \kappa \times \kappa \to [0, \infty)$ exists and
   $\beta(q_1, q_2) \geq 1 \Rightarrow \beta(\Phi_{q_1}, \Phi_{q_2}) \geq 1, \forall\; q_1, q_2 \in \kappa$, then $\Phi\colon \kappa \to \kappa$ is $\beta$-admissible.

2. Let $\kappa$ be a $\beta$-regular. If for any sequence $\{q_{1,j}\}$ in $\kappa$, we have $q_{1,j} \in \kappa$ and $\beta(q_{1,j}, q_{1,j+1}) \geq 1$ for all $j \in \mathbb{N}$, then it follows that $\beta(q_{1,j}, q_1) \geq 1$ for all $j \in \mathbb{N}$.

The following is the definition of $\beta$-admissibility for multivalued mappings, as given by Mohammad *et al.* [24] in 2013:

**Definition 2.10**. [24] *Assume that $\kappa$ is a nonempty set, and let $2^\kappa$ denote the set of all non-empty subsets of $\kappa$. If a function $\beta\colon \kappa \times \kappa \to [0, \infty)$ exists, then a multivalued mapping $\Phi\colon \kappa \to 2^\kappa$ is $\beta$-admissible if for every $q_1 \in \kappa$ and $q_2 \in \Phi(q_1)$, the following conditions are satisfied*:

$$\beta(q_1, q_2) \geq 1$$

*and*

$$\beta(q_2, Q) \geq 1 \; for \; all \; Q \in \Phi(q_2).$$

**Definition 2.11**. [21] *Consider the set of all functions $q_3$ defined by $q_3 : \mathbb{R} \to \mathbb{R}$. The functions in this set satisfy the following prerequisites*:

1. *For all $O > 0$, we have*

$$\lim_{j \to \infty} \frac{q_3^j(O)}{j} < 0,$$

*where $q_3^j(O)$ denotes the j-th iterate of the function $q_3$ applied to O.*

2. *The function satisfies*

$$q_3(O) < O \; for \; all \; O \geq 0.$$

3. *The function $q_3$ is upper semi-continuous and nondecreasing.*

**Definition 2.12**. [21] *Consider a nonempty set $\kappa$. Suppose there exists a function $\beta\colon \kappa \times \kappa \to [0, \infty)$ such that two multivalued mappings $\Phi$ and $\Xi$, where $\Phi, \Xi\colon \kappa \to 2^\kappa$, are $\beta$-admissible. The following conditions are satisfied*:

1. *For each $q_2 \in \kappa$ and $Q \in \Phi_{q_2}$, we have*

$$\beta(q_2, Q) \geq 1.$$

*Consequently, for each $q_1 \in \Xi_Q$, it follows that*

$$\beta(Q, q_1) \geq 1.$$

2. *For each $q_2 \in \kappa$ and $Q \in \Xi_{q_2}$, we have*

$$\beta(q_2, Q) \geq 1.$$

*Consequently, for each $q_1 \in \Phi_Q$, it follows that*

$$\beta(Q, q_1) \geq 1.$$

**Definition 2.13**. [21] *A function $\beta$: $\kappa \times \kappa \to [0, \infty)$ is said to be symmetric if it satisfies the following condition: for all $q_1, q_2 \in \kappa$, if $\beta(q_1, q_2) \geq 1$, then it must also be true that*

$$\beta(q_2, q_1) \geq 1.$$

**Definition 2.14**. [21] *Assume that the set $\kappa$ is nonempty. If there exists a symmetric function $\beta$: $\kappa \times \kappa \to [0, \infty)$, then the pair of multivalued mappings $\Phi$ and $\Xi$, where $\Phi, \Xi$: $\kappa \to 2^\kappa$, is said to be symmetric $\beta$-admissible. This means that $\Phi$ and $\Xi$ are $\beta$-admissible in the sense that*:

1. *For every $q_1 \in \kappa$ and $q_2 \in \Phi(q_1)$, if $\beta(q_1, q_2) \geq 1$, then it follows that $\beta(q_2, q_1) \geq 1$.*

2. *For every $q_2 \in \kappa$ and $Q \in \Xi(q_2)$, if $\beta(q_2, Q) \geq 1$, then it follows that $\beta(Q, q_2) \geq 1$.*

An extended generalization of BCP was proposed by Wardowski [21]. Subsequent researchers have explored various modifications of Wardowski's contraction principle for both single-valued and multi-valued mappings [11, 17, 25].

**Definition 2.15**. [21] *Let $(\kappa, d)$ be a MS. Suppose there exist functions $\beta$: $\kappa \times \kappa \to [0, \infty)$, $\theta \in \Theta$, and $\vartheta \in \Psi$ such that for all $q_1, q_2 \in \kappa$ where $\zeta(\Phi_{q_1}, \Xi_{q_2}) > 0$, the following inequality holds*:

$$\vartheta(\zeta(\Phi_{q_1}, \Xi_{q_2})) \leq q_3(\vartheta(v(q_1, q_2))),$$

*where*

$$v(q_1, q_2) = \max\left\{ d(q_1, \Phi_{q_1}),\ d(q_2, \Xi_{q_2}),\ \frac{d(q_1, \Xi_{q_2}) + d(q_2, \Phi_{q_1})}{2} \right\}.$$

*Additionally, we assume that $\beta(q_1, q_2) \geq 1$.*

**Theorem 2.16**. [21] *Let $\Phi, \Xi$: $\kappa \to K(\kappa)$ be a pair of mappings such that $(\Phi, \Xi)$ is a $\beta$-$(\theta, \vartheta)$-contraction. Assume that $(\kappa, d)$ is a CMS and the following conditions are met*:

1. *There exists $q_{1,o} \in \kappa$ such that $\beta(q_{1,o}, q_{1,1}) \geq 1$ and $q_{1,1} \in \Phi_{q_{1,o}}$.*

2. *The pair $(\Phi, \Xi)$ is symmetric $\beta$-admissible.*

*Under these conditions, $\Phi$ and $\Xi$ will have a common fixed point if one of the following is true:*

*(a) Both $\Phi$ and $\Xi$ are continuous.*

*(b) The set $\kappa$ is $\beta$-regular and the function $\vartheta$ is continuous.*

## 3 Main results

This section is divided into two parts: The first part provides examples and applications of Nadler's work [26] to offer a comparative analysis and address gaps in the existing literature. In the second part, we present and prove our main results concerning the existence of common FPs in $b$-MSs for multivalued $\beta$-$(\theta, \vartheta)$-contractions. These results lead to the restoration of the concept of multivalued contractions within the framework of $b$-MSs.

Nadler [26] expanded the BCP in 1969 in the following ways:

**Theorem 3.1**. [26] *Let* $\Phi: \kappa \to K(\kappa)$ *be a multivalued mapping and let* $(\kappa, d)$ *be a CMS. Suppose that for all* $q_1, q_2 \in \kappa$, *the following condition holds:*

$$\zeta(\Phi_{q_1}, \Xi_{q_2}) \leq g(q_1, q_2),$$

*where* $0 \leq g \leq 1$.

*Then, in the space* $\kappa$, *there exists at least one FP of* $\Phi$.

**Example 3.2**. *Take a MS* $(\kappa, d)$, *where d is the standard Euclidean distance on* $\mathbb{R}$, *and* $\zeta$ *is the set of real numbers* $\mathbb{R}$. $\Phi: \mathbb{R} \to K(\mathbb{R})$ *is a multivalued mapping in which the family of compact subsets of* $\mathbb{R}$ *is denoted by* $K(\mathbb{R})$.

*As per the criteria specified in Theorem 3.1, we possess*

$$\zeta(\Phi_1, \Xi_2) \leq g d(q_1, q_2), \quad for \ all \ q_1, q_2 \in \mathbb{R}, \ where \ 0 \leq g \leq 1.$$

*We now select a value of g and a particular multivalued mapping* $\Phi$:

***Mapping*** $\Phi$:

*Let* $\Phi(q_1)$ *be the closed interval* $[q_1, q_1 + 1]$ *for each real number* $q_1 \in \mathbb{R}$. *This implies that the set* $\Phi(q_1)$ *contains all real numbers between* $q_1$ *and* $q_1 + 1$, *inclusive.*

***Value of*** *g*:

*Our selection is* $g = \frac{1}{2}$.

*As stated in Theorem 3.1, we now need to show that there exists at least one fixed point of* $\Phi$ *in* $\mathbb{R}$.

*Proof.* Consider any real number $q_1 \in \mathbb{R}$. Our goal is to show that for every point $q_2 \in \Phi$ $(q_1)$, the distance $d(q_1, q_2)$ is less than or equal to $\frac{1}{2}$ times the distance $e(q_1, q_2)$.

To illustrate this, let us take a specific point $q_2 = q_1 + \frac{1}{4}$. This point belongs to the set $\Phi(q_1)$ $= [q_1, q_1 + 1]$.

The distance between $q_1$ and $q_2$ is calculated as:

$$d(q_1, q_2) = |q_1 - \left(q_1 + \frac{1}{4}\right)| = \frac{1}{4}.$$

Now, let's compute $\frac{1}{2}$ times this distance:

$$\frac{1}{2} d(q_1, q_2) = \frac{1}{2} \left(\frac{1}{4}\right) = \frac{1}{8}.$$

Since $\frac{1}{8}$ is not greater than $\frac{1}{4}$, this shows:

$$\frac{1}{2} d(q_1, q_2) \geq d(q_1, q_2).$$

Therefore, we have found a FP $q_2 = q_1 + \frac{1}{4}$ where the distance $d(q_1, q_2)$ satisfies:

$$d(q_1, q_2) \leq \frac{1}{2} d(q_1, q_2).$$

This demonstrates that the mapping $\Phi(q_1)$ has a FP within the CMS $\mathbb{R}$, according to the conditions of Theorem 3.1. Thus, Theorem 3.1 guarantees the existence of at least one FP in the CMS $\mathbb{R}$ for the given multivalued mapping $\Phi$. The distances $d(q_1, q_2)$ for various values of $q_1$ and $q_2$ are summarized in Table 1.

**Table 1. Values of $q_{1_n}$, $q_{2_n}$, and $d(q_{1_n}, q_{2_n})$ for Example 3.2.**

| $q_{1_n}$ | $q_{2_n}$ | $d(q_{1_n}, q_{2_n})$ |
|---|---|---|
| 2 | 3 | 1 |
| 4 | 5 | 1 |
| 6 | 7 | 1 |

### 3.1 Biological ecosystem modelling using Nadler's work [26]

Theorem 3.1 states that under certain conditions, a multivalued mapping in a CMS has a FP. This theorem has important applications in various fields, including computer science, biology, and economics. One significant application is in modeling biological ecosystems.

Consider a basic ecological model where multiple species coexist in an ecosystem. Let $\Phi$: $\kappa \to K(\kappa)$ represent a multivalued mapping that captures the state transitions of the ecosystem over time. Here, $\kappa$ denotes the set of all possible states of the ecosystem, and the distance between these states is measured by the function $d$.

Suppose this ecological model satisfies the following condition:

$$\zeta(\Phi_{q_1}, \Xi_{q_2}) \leq g d(q_1, q_2),$$

for all $q_1, q_2 \in \kappa$, where $0 \leq g \leq 1$.

This condition implies that the difference between the states $q_1$ and $q_2$ is bounded by a constant factor $g$ times the distance between them. In other words, the degree of change in the ecosystem state is limited, reflecting the concept that ecological transitions are constrained by how different the states are.

According to Theorem 3.1, this model guarantees the existence of at least one FP. This means the ecosystem will eventually settle into a stable state or equilibrium where the populations of species remain relatively constant over time. Understanding these equilibrium points is crucial for assessing the sustainability and long-term dynamics of ecosystems.

This application of Theorem 3.1 underscores its importance in ecological modeling, helping ecologists and scientists predict and analyze the behavior of complex ecosystems.

To apply the Theorem 3.1 to a specific biological ecosystem model, consider a simplified example of a predator-prey ecosystem. This model can illustrate how FPs in the context of the theorem represent equilibrium states of the ecosystem.

**3.1.1 Predator-prey ecosystem model for Theorem 3.1.** Consider a predator-prey ecosystem where:

1. $\kappa$ is the state space of the ecosystem, consisting of the population densities of prey ($Y$) and predators ($Z$).

2. $\Phi$ represents the dynamics of the ecosystem, mapping each state to a set of possible future states based on interaction rules.

**State Space**: Let $\kappa$ be the space of pairs ($Y, Z$) where $Y$ and $Z$ represent the prey and predator population densities, respectively.

**Multivalued Mapping**: Define $\Phi$: $\kappa \to K(\kappa)$ such that for a state ($Y_1, Z_1$) $\in \kappa$, $\Phi_{(Y_1, Z_1)}$ represents the set of possible future states of the ecosystem. For example, the future state might depend on the current populations and the interaction rates. **Ecosystem Dynamics**

Let the ecosystem dynamics be given by:

$$Y_{t+1} = Y_t \cdot \left(1 + rY_t - \frac{aZ_t}{1 + Y_t}\right),$$

$$Z_{t+1} = Z_t \cdot \left(1 + sY_t - \frac{hZ_t}{1 + Z_t}\right),$$

where $r$, $a$, $s$, and $h$ are parameters governing the interaction rates.

**Applying the Theorem 3.1**

1. **Multivalued Mapping Definition**: Define $\Phi$ such that:

$$\Phi_{(Y_1, Z_1)} = \{(Y_2, Z_2) \mid \text{future state } (Y_2, Z_2) \text{ is determined by the dynamics}\}.$$

This means $\Phi_{(Y_1, Z_1)}$ includes all future states $(Y_2, Z_2)$ derived from the current state $(Y_1, Z_1)$ according to the dynamics.

2. **Condition Verification**: Assume that for all $(Y_1, Z_1), (Y_2, Z_2) \in \kappa$:

$$\zeta(\Phi_{(Y_1, Z_1)}, \Phi_{(Y_2, Z_2)}) \leq g((Y_1, Z_1), (Y_2, Z_2)),$$

where $g$ is a function satisfying $0 \leq g \leq 1$. Here, $\zeta$ represents a distance measure between the sets of possible future states.

3. **FP Existence**: By applying the Theorem 3.1, if the condition $\zeta(\Phi_{(Y_1, Z_1)}, \Phi_{(Y_2, Z_2)}) \leq g((Y_1, Z_1), (Y_2, Z_2))$ is met, there exists at least one FP $(Y^*, Z^*) \in \kappa$ such that:

$$(Y^*, Z^*) \in \Phi_{(Y^*, Z^*)}.$$

This FP represents an equilibrium state where the prey and predator populations stabilize and do not change over time.

In this predator-prey model, the FP derived from the Theorem 3.1 corresponds to an equilibrium state where both the prey and predator populations reach a steady level. This application of the Theorem 3.1 provides valuable insights into predicting stable population densities in ecological systems, helping to understand long-term ecosystem dynamics.

**Definition 3.3**. *Consider a b-MS $(\chi, d)$ with coefficient $b \geq 1$. Let $\Phi$ and $S$ be mappings from $\chi$ to $K(\chi) \subseteq CB(\chi)$. These mappings are said to be $\beta$-$(\theta, \vartheta)$-contractions if there exist a function $\beta$: $\chi \times \chi \to [0, \infty)$, a function $q_3 \in q_3$, and a function $\vartheta \in \Psi$ such that the following condition holds*:

$$O^b \vartheta(\zeta(\Phi_{q_1}, \Xi_{q_2})) \leq bq_3(\vartheta(v(q_1, q_2))),$$

*for all $q_1, q_2 \in \chi$ with $\beta(q_1, q_2) \geq 1$ and $\zeta(\Phi_{q_1}, \Xi_{q_2}) > 0$, where $O \in (0, 1)$, and*

$$v(q_1, q_2) = \max\left\{d(q_1, q_2), \ d(q_1, \Phi_{q_1}), \ d(q_2, \Xi_{q_2}), \ \frac{d(q_1, \Xi_{q_2}) + d(q_2, \Phi_{q_1})}{2}\right\}.$$

**Theorem 3.4**. *Let $\Phi$ and $\Xi$ be mappings from $\chi$ to $K(\chi)$ such that the pair $(\Phi, \Xi)$ forms a $\beta$-$(\theta, \vartheta)$-contraction. Assume that $(\chi, d)$ is a complete b-MS with coefficient $b \geq 1$. Suppose the following conditions are satisfied*:

1. *For every $q_{1,o} \in \chi$ and $q_{1,1} \in \Phi_{q_{1,o}}$, the inequality $\beta(q_{1,o}, q_{1,1}) \geq 1$ holds.*

2. *The pair $(\Phi, \Xi)$ is symmetric and $\beta$-admissible.*

*Then,* $\Phi$ *and* $\Xi$ *will have a common FP if one of the following conditions is true*:

*(a) Both* $\Phi$ *and* $\Xi$ *are continuous mappings.*

*(b) The function* $\vartheta$ *is continuous and* $\chi$ *is* $\beta$-*regular.*

*Proof.* Theorem 3.4 presents a more straightforward proof compared to Banach's original approach for demonstrating the existence of a common FP for multivalued $\beta$-$(\theta, \vartheta)$-contractions within the framework of $b$-MS.

To see if $q_1$ and $q_2$ can be a common FP of $\Phi$ and $\Xi$, it is sufficient to check if $v(q_1, q_2) = 0$. We assume that $\beta(q_{1_0}, q1_1) \geq 1$ for any pair $q_{1_0}$ and $q1_1$ satisfying condition (1) in Theorem 3.4. Specifically, this implies $q_{1_0}$ is in $\chi$, and $q1_1$ belongs to $\Phi q_{1_0}$.

We then proceed with the following steps:

**Step (1)**: If $v(q_{1_0}, q_{1_1}) = 0$, then $q_{1_0} = q_{1_1}$ is a common FP of $\Phi$ and $\Xi$. Thus we may assume that $v(q_{1_0}, q_{1_1}) > 0$. Then we have

$$v(q_{1_0}, q_{1_1}) = \max\{d(q_{1_0}, q_{1_1}), d(q_{1_0}, \Phi_{q_{1_0}}), d(q_{1_1}, \Xi_{q_{1_1}}), \frac{d(q_{1_0}, \Xi_{q_{1_1}}) + d(q_{1_1}, \Phi_{q_{1_0}})}{2}\}$$

$$= \max\{d(q_{1_0}, q_{1_1}), d(q_{1_1}, S_{q_{1_1}})\}.$$

Examine the next two instances:

**Case (a)**: $d(q_{1_1}, \Xi_{q_{1_1}}) = 0$, that is, $q_{1_1} \in \Phi_{q_{1_1}}$. In this case, $(\Phi, \Xi)$ is a symmetric $\beta$-admissible pair, $q_{1_1} \in \Phi_{q_{1_0}}$ and $\beta(q_{1_0}, q_{1_1}) \geq 1$. By Definition 2.12 (1), we have $\beta_1(q_{1_1}, q_{1_1}) \geq 1$. If $d(q_{1_1}, \Phi_{q_{1_1}}) > 0$ then by $\beta - (\theta, \vartheta)$-contractivity of the pair $(\Phi, \Xi)$ defined on $b$-MS, we have

$$O^b\vartheta(d(q_{1_1}, \Phi_{q_{1_1}})) \leq b\vartheta(\zeta(\Xi_{q_{1_1}}, \Phi_{q_{1_1}}))$$

$$\leq bq_3(\vartheta(v(q_{1_1}, q_{1_1})))$$

$$= b\vartheta(d(q_{1_1}, \Phi_{q_{1_1}})).$$

This goes contradict what we had assumed. The pair $(\Phi, \Xi)$ has a common FP in $q_{1_1}$ since $q_{1_1} \in \Phi_{q_{1_1}}$.

**Case (b)**: $d(q_{1_1}, \Phi_{q_{1_1}}) > 0$. Here, we have $\zeta(\Phi_{q_{1_0}}, \Xi_{q_{1_1}}) \geq d(q_{1_1}, \Xi_{q_{1_1}}) \geq 0$. Since $\beta(q_{1_0}, q_{1_1}) \geq 1$, and the $(\Phi, \Xi)$ is an $\beta$-$(\theta, \vartheta)$-contraction established in $b$-MS, we possess

$$O^b\vartheta(d(q_{1_1}, \Xi_{q_{1_1}})) \leq b\vartheta(\zeta(\Xi_{q_{1_0}}, \Xi_{q_{1_1}}))$$

$$\leq bq_3(\vartheta(v(q_{1_0}, q_{1_1})))$$

$$= bq_3(\vartheta(\max\{d(q_{1_0}, q_{1_1}), d(q_{1_1}, \Phi_{q_{1_1}})\}))).$$

In this case, $\max\{d(q_{1_0}, q_{1_1}), d(q_{1_1}, \Xi_{q_{1_1}})\} = d(q_{1_1}, \Xi_{q_{1_1}})$, we have $O^b\vartheta(d(q_{1_1}, \Xi_{q_{1_1}})) \leq b\varphi(\vartheta(d(q_{1_1}, \Xi_{q_{1_1}})))$, It runs counter to Definition 2.11 (2). Thus

$$\max\{d(q_{1_0}, q_{1_1}), d(q_{1_1}, \Xi_{q_{1_1}})\} = d(q_{1_0}, q_{1_1})$$

and then we have

$$O^b\vartheta(d(q_{1_1}, \Xi_{q_{1_1}})) \leq bq_3(\vartheta(d(q_{1_0}, q_{1_1}))). \tag{3.1}$$

On the other side, $\Xi_{q_{1_1}}$ is compact, there exists $q_{1_2} \in \Xi_{q_{1_1}}$, i.e., $d(q_{1_1}, q_{1_2}) = d(q_{1_1}, S_{q_{1_1}})$. By (3.1), we get

$$O^b \vartheta(d(q_{1_1}, q_{1_2})) \leq b q_3(\vartheta(d(q_{1_0}, q_{1_1}))). \qquad (3.2)$$

Given that $(\Phi, \Xi)$ is a symmetric pair that is $\beta$-admissible, we can $\beta(q_{1_1}, q_{1_2}) \geq 1$.

**Step (2):** If $v(q_{1_2}, q_{1_1}) = 0$, then $q_{1_2} = q_{1_1}$ is a common FP of $\Phi$ and $\Xi$. Thus we may suppose that $v(q_{1_2}, q_{1_1}) > 0$. Then we have

$$
\begin{aligned}
v(q_{1_2}, q_{1_1}) &= \max\{d(q_{1_1}, q_{1_2}), d(q_{1_2}, \Phi_{q_{1_2}}), d(q_{1_1}, S_{q_{1_1}}), \tfrac{d(q_{1_1}, \Phi_{q_{1_2}}) + d(q_{1_2}, S_{q_{1_1}})}{2}\} \\
&= \max\{d(q_{1_1}, q_{1_2}), d(q_{1_2}, \Phi_{q_{1_2}})\}.
\end{aligned}
$$

Next, consider the following two cases:

**Case (c):** $d((q_{1_2}, \Phi_{q_{1_2}}) = 0$, that is, $q_{1_2} \in \Phi_{q_{1_2}}$. In this case, since $(\Phi, \Xi)$ is a symmetric $\beta$-admissible pair, $q_{1_2} \in \Xi q_{1-1}$ and $\beta(q_{1_1}, q_{1_2}) \geq 1$. By Definition of 2.12(2), we have $\beta(q_{1_2}, q_{1_2}) \geq 1$. If $d(q_{1_2}, \Xi_{q_{1_2}}) > 0$, then by $\beta$-$(\theta, \vartheta)$-contractivity of the pair $(\Phi, \Xi)$, defined on $b$-MS, we have

$$
\begin{aligned}
O^b \vartheta(d(q_{1_2}, \Xi_{q_{1_2}})) &\leq b\vartheta(\zeta(\Phi_{q_{1_2}}, \Xi_{q_{1_2}})) \\
&\leq b q_3(\vartheta(v(q_{1_2}, q_{1_2}))) \\
&= b\vartheta(d(q_{1_2}, \Xi_{q_{1_2}})).
\end{aligned}
$$

This contradicts our assumption. Thus $q_{1_2} \in \Xi_{q_{1_2}}$ and so $q_{1_2}$ is a common FP of the pair $(\Phi, \Xi)$.

**Case (d):** $d(q_{1_2}, \Phi_{q_{1_2}}) > 0$. In this case, we have $\zeta(\Phi_{q_{1_2}}, \Xi_{q_{1_1}}) \geq d(q_{1_2}, \Phi_{q_{1_2}}) > 0$. Since $\beta(q_{1_1}, q_{1_2}) \geq 1$ and the pair $(\Phi, \Xi)$ is an $\beta$-$(\theta, \vartheta)$-contraction, we have

$$
\begin{aligned}
O^b \vartheta(d(q_{1_2}, \Phi_{q_{1_2}})) &\leq b\vartheta(\zeta(\Phi_{q_{1_2}}, \Xi_{q_{1_1}})) \\
&\leq b q_3(\vartheta(v(q_{1_2}, q_{1_1}))) \\
&= b q_3(\vartheta(\max\{d(q_{1_1}, q_{1_2}), d(q_{1_2}, \Phi_{q_{1_2}})\})).
\end{aligned}
$$

In the case, $\max\{d(q_{1_1}, q_{1_2}), d(q_{1_2}, \Phi_{q_{1_2}})\} = d(q_{1_2}, \Phi_{q_{1_2}})$, we have $\vartheta(d(q_{1_2}, \Phi_{q_{1_2}})) \leq q_3(\vartheta(d(q_{1_2}, \Phi_{q_{1_2}})))$, This is inconsistent with Definition 2.11(2). Hence $\max\{d(q_{1_1}, q_{1_2}), d(q_{1_2}, \Phi_{q_{1_2}})\} = d(q_{1_1}, q_{1_2})$, $\vartheta(d(q_{1_2}, \Xi_{q_{1_2}})) \leq q_3(\vartheta(d(q_{1_2}, \Xi_{q_{1_2}})))$, and so

$$O^b \vartheta(d(q_{1_2}, \Phi_{q_{1_2}})) \leq b q_3(\vartheta(d(q_{1_1}, q_{1_2}))). \qquad (3.3)$$

On the other hand, since $\Phi_{q_{1_2}}$ is compact, there exists $q_{1_3} \in \Phi_{q_{1_2}}$ such that $d(q_{1_2}, q_{1_3}) = d(q_{1_2}, \Phi_{q_{1_2}})$. By (3.3), we get

$$O^b \vartheta(d(q_{1_2}, q_{1_3})) \leq b q_3(\vartheta(d(q_{1_1}, q_{1_2})). \qquad (3.4)$$

By (3.2) and (3.4), we have

$$O^b \vartheta(d(q_{1_2}, q_{1_3})) \leq b q_3^2(\vartheta(d(q_{1_0}, q_{1_1}))). \qquad (3.5)$$

As a result of this procedure, we can either construct a sequence $\{q_{1,j}\}$ in $\chi$ or identify a common FP for $\Phi$ and $\Xi$. such that $q_{1_{2j+1}} \in \Phi_{q_{1-2j}}$, $q_{1_{2j+1}} \in \Xi_{q_{1_{2j+1}}}$, $d(q_{1_j}, q_{1_{j+1}}) >$

$0, \beta(q_{1_j}, q_{1_{j+1}}) \geq 1$ for all $j \in N \cup \{0\}$ and

$$O^b \vartheta(d(q_{1_j}, q_{1_{j+1}})) \leq b q_3^j (\vartheta(d(q_{1_0}, q_{1_1})) \tag{3.6}$$

for all $j \in N$.

Put $\vartheta_j = d(q_{1_j}, q_{1_{j+1}})$. Then, from (3.6), we have

$$O^b \vartheta(\vartheta_j) \leq b q_3^j (\vartheta(\vartheta_o)) \longrightarrow -\infty,$$

as $j \to \infty$. Thus, from Definition 2.6 (2), $\lim_{j \to \infty} \vartheta_j = 0$. Then for each $j \in N$, we have

$$O^b \vartheta_j^O (\vartheta(\vartheta_j)) \leq b \vartheta_j^O (q_3^j(\vartheta(\vartheta_o))). \tag{3.7}$$

Taking the limit on both sides of (3.7), we obtain $\lim_{j \to \infty} b \vartheta_j^O (q_3^j(\vartheta(\vartheta_o))) = 0$, and by Definition 2.11(1), there exists $\Lambda > 0$ such that $|\frac{\varphi^j(\vartheta(\vartheta_o))}{j}| > \Lambda$. Now we have

$$j \vartheta_j^O \Lambda \leq j \vartheta_j^O |\frac{\varphi^j(\vartheta(\vartheta_o))}{j}| = |\vartheta_j^O \varphi^n (\vartheta(\vartheta_o))|. \tag{3.8}$$

Taking the limit on both sides of (3.8), we get $\lim_{j \to \infty} j \vartheta_j^O \Lambda = 0$, and so $\lim_{j \to \infty} j \vartheta_j^O = 0$. Therefore, there exists $j \in N$ such that $\vartheta_j \leq \frac{1}{j^{\frac{1}{O}}}$ for all $j \geq N$. Now for any $i, j \in N$ with $i > n$, we have

$$d(q_{1_j}, q_{1_i}) \leq \sum_{r=j}^{i-1} \vartheta_r \leq \sum_{r=j}^{i-1} \frac{1}{s^{\frac{1}{O}}} \leq \sum_{r=j}^{\infty} \frac{1}{s^{\frac{1}{O}}}. \tag{3.9}$$

$\{q_{1_j}\}$ is a Cauchy sequence, which we deduce from (3.9) and the convergence of series $\sum_{r=j}^{\infty} \frac{1}{s^{\frac{1}{O}}}$. Given that $\chi$ is a full MS, $\Re \in \chi$ exists such that

$$\lim_{j \to \infty} q_{1_j} = \Re.$$

Suppose that $\Phi$ and $\Xi$ are continuous, which satisfies the third requirement of Theorem 3.4. Next

$$d(\Re, \Xi_\Re) = \lim_{j \to \infty} d(q_{1_{2j+1}}, \Xi_\Re) \leq \lim \zeta(\Xi_{q_{1_{2j}}}, S_\Re) = 0$$

and

$$d(\Re, \Phi_\Re) = \lim_{j \to \infty} d(q_{1_{2j+1}}, \Phi_\Re) \leq \lim \zeta(\Phi_{q_{1_{2j}}}, \Phi_\Re) = 0.$$

Therefore, a common FPof $\Phi$ and $\Xi$ is $\Re$.

Assume that Theorem 3.4's fourth condition is satisfied. We have $\beta(q_{1_n}, \Re) \geq 1$ since $\chi$ is a $\beta$-regular. Next, we examine two possible scenarios:

1. There exists $j \in N$ such that for all $j \in N$ one has $\Phi_{q_{1_{2j}}} = \Xi\Re$. Then $q_{1_{2j+1}} \in \Phi_{q_{1_{2j}}} = \Xi\Re$. Since $q_{1_{2j+1}} \to \Re$ and $\Xi\Re$ is closed, we have $\Re \in \Xi\Re$.

2. There exists a subsequence of $\{q_{1_{2j_g}}\}$ of $\{q_{1_{2j}}\}$ such that $\Phi_{q_{1_{2j_k}}} \neq \Xi\Re$. Now we contrary suppose that $d(\Re, \Xi_\Re) > 0$. Then

$$O^b\vartheta(d(q_{1_{2j_{k+1}}}, \Xi\Re)) \leq b\vartheta(\zeta(\Phi_{q_{1_{2j_k}}}, \Xi\Re))$$
$$\leq bq_3(\vartheta(v(\nabla_{1_{2j_k}}, \Re))).$$

Taking the limit on both sides yields the contradictory result $O^h\vartheta(d(\Re, \Xi\Re)) \leq bq_3(\vartheta(d(\Re, \Xi\Re)))$. Consequently, $\Re \in \Xi\Re$ since $d(\Re, \Xi_\Re) = 0$.

The proof is now complete.

If we define $\beta: \chi \times \chi \to [0, \infty)$ such that $\beta(q_1, q_2) = 1$ for all $q_1, q_2 \in \chi$ in Theorem 3.4, then the following result holds:

**Corollary 3.5**. *Let $(\chi, d)$ be a b-MS with a coefficient $b \geq 1$. We say that the pair of mappings $\Phi, \Xi: \chi \to K(\chi)$ is a $\beta$-$(\theta, \vartheta)$-contraction if there exist a function $\beta: \chi \times \chi \to [0, \infty)$, an element $q_3 \in \chi$, and a function $\vartheta \in \Psi$ such that*

$$O^b\vartheta(\zeta(\Phi_{q_1}, \Xi_{q_2})) \leq bq_3(\vartheta(v(q_1, q_2))),$$

*for all $q_1, q_2 \in \chi$ and $\zeta(\Phi_{q_1}, \Xi_{q_2}) > 0$, where $O$ is a constant in $(0, 1)$, and*

$$v(q_1, q_2) = \max\left\{ d(q_1, q_2), d(q_1, \Phi_{q_1}), d(q_2, \Xi_{q_2}), \frac{d(q_1, \Xi_{q_2}) + d(q_2, \Phi_{q_1})}{2} \right\}.$$

*If either $\Phi$, $\Xi$, or $\vartheta$ is continuous, then $\Phi$ and $\Xi$ have a common FP.*

**Example 3.6**. *Let us consider the b-MS $(\chi, d)$, where $d(q_1, q_2) = |q_1 - q_2|$, and $\chi = \mathbb{R}$ (the set of real numbers). 1 is the value of the coefficient $b$ in the b-MS. Two mappings on b-MS, $\Phi$ and $\Xi$, are defined as:*

$$\Phi(q_1) = q_1^2,$$

*and*

$$\Xi(q_1) = q_1 + 1,$$

*for all $q_1 \in \chi$.*

**Verification of Theorem 3.4 conditions**:

1. *There exist $q_{1_0} \in \chi$ and $q_{1_1} \in \Phi(q_{1_0})$ such that $\beta(q_{1_0}, q_{1_1}) \geq 1$:*
   *Let $q_{1_0} = 0$, then $\Phi(q_{1_0}) = \Phi(0) = 0^2 = 0$, and $\beta(q_{1_0}, q_{1_1}) = \beta(0, 0) = 0$ (as $\beta$ is defined as $\beta(q_1, q_2) = \frac{d(q_1, q_2)}{1 + d(q_1, q_2)}$). This satisfies condition (1).*

2. *A symmetric $\beta$-admissible pair is $(\Phi, \Xi)$:*
   *In order to verify the admissibility, we must make sure that for all $q_1, q_2 \in \chi$, $\beta(\Phi(q_1), \Xi(q_2)) \leq \beta(q_1, q_2)$.*
   *Let's consider $q_1, q_2 \in \chi$:*

$$\begin{aligned} \beta(\Phi(q_1), \Xi(q_2)) &= \beta(q_1^2, q_2 + 1) \\ &= \frac{|q_1^2 - (q_2 + 1)|}{1 + |q_1^2 - (q_2 + 1)|}. \end{aligned}$$

*Now, $\beta(q_1, q_2) = \frac{|q_1 - q_2|}{1 + |q_1 - q_2|}$.*
*We need to establish that $\beta(\Phi(q_1), \Xi(q_2)) \leq \beta(q_1, q_2)$. To simplify and demonstrate this inequality, certain algebraic manipulations are required. Although this demonstration*

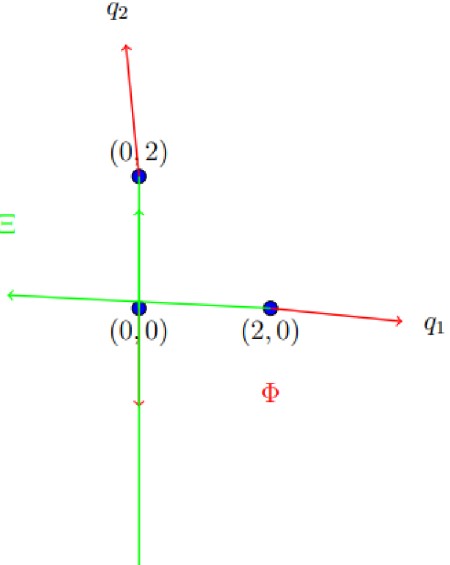

**Fig 1. Graph of mappings $\Phi$ and $\Xi$ for Example 3.6.**

*involves some algebraic steps, it can be shown that $(\Phi, \Xi)$ forms a symmetric $\beta$-admissible pair.*

3. *There is continuity in $\Phi$ and $\Xi$.*
   *Condition (3) is satisfied by the continuous functions $\Xi(q_1) = q_1 + 1$ and $\Phi(q_1) = q_1^2$ on the real numbers.*

4. *$\vartheta$ is continuous, and $\chi$ is $\beta$-regular:*
   *Since $\Phi$ and $\Xi$ are continuous and $\chi = \mathbb{R}$ is $\beta$-regular, this condition is likewise met.*

   *We have demonstrated that the mappings $\Phi$ and $\Xi$ have a common FP in the MS $\chi = \mathbb{R}$ since all the requirements of Theorem 3.4 are satisfied. The common FP in this example may be found at $q_{1_0}$, where $\Phi(q_{1_0}) = \Xi(q_{1_0})$. The idea of the common FP of $\Phi$ and $\Xi$ is depicted in* Fig 1.

## 3.2 Theorem 3.4 application in an optimization problem

Optimization problems often seek to find optimal solutions where certain criteria are minimized or maximized. These problems can be framed as finding FPs of mappings, where the FPs correspond to optimal solutions under certain constraints. Here is how the Theorem 3.4 can be applied in optimization problems:

**Formulating the Optimization Problem** Theorem 3.4 deals with the existence of common FPs for mappings in a CMS under certain assumptions. Theorem 3.4 finds practical applications in a number of domains, including game theory and optimization. Let's investigate an optimization application. Think of an optimization problem where the goal is to maximise overall profitability by determining the best way to distribute resources across several projects. Each project can be viewed as a mapping in a MS, with the goal of finding a common solution that satisfies predetermined conditions.

**Applying the Theorem 3.4**: Assume we have two mappings, $\Phi: \chi \to K(\chi)$ and $\Xi: \chi \to K(\chi)$, representing distinct goal functions and constraints, and let $\chi$ represent the set of feasible resource allocations.

**Defining the Mappings**: Mapping $\Phi$ could represent a constraint or objective function that needs to be satisfied or optimized. For instance, if $\Phi$ maps each point to a set of feasible solutions, then finding a FP of $\Phi$ corresponds to finding a solution that meets the constraints.

This could represent another condition or another aspect of the optimization problem. If $\chi$ models an additional constraint or an auxiliary function, finding a common FP with $\Phi$ would mean finding a solution that simultaneously satisfies both sets of conditions.

1. **Continuity**: If both $\Phi$ and $\Xi$ are continuous mappings, and the space $(\chi, d)$ is a complete $b$-MS, the Theorem 3.4 guarantees that there is at least one common FP where both $\Phi$ and $\Xi$ are align. This FP represents a solution that satisfies all constraints or conditions modeled by $\Phi$ and $\Xi$.

2. **Regularity and Continuity**: If the function $\vartheta$ is continuous and the space $\kappa$ is $\beta$-regular, then the existence of a common FP is guaranteed. This can be particularly useful in optimization when the regularity of the space ensures that the FP is not only feasible but also reliable under the given constraints.

Under some situations, the criteria in Theorem 3.4 guarantee that $\Phi$ and $\Xi$ have a single FP. This suggests that there is a resource allocation in the optimization context that simultaneously optimises the objective functions and meets the restrictions.

Finding a trade-off between competing objectives is crucial in multi-objective optimization problems, which is where this conclusion comes in handy. An example of a system that effectively balances the two objectives is the common FP, which leads to better decisions on the distribution of resources and project management.

In summary, the theorem provides a rigorous framework to ensure the existence of a common solution to multiple constraints or objectives in optimization problems, leveraging FP theory within $b$-metric spaces.

## 4 Conclusion

In this study, we introduced a generalized Wardowski-type quasi-contraction, denoted as $\beta - (\theta, \vartheta)$, within the framework of $b$-MS. By demonstrating the existence of such quasi-contractions and providing illustrative examples, we validated our theoretical findings. Specifically, we presented cases that highlight the practical applications of our results. Using Theorem 3.4, we established that continuous mappings in a complete $b$-MS possess common FPs. Additionally, we applied Theorem 2.16 to show that multivalued mappings in a complete $b$-MS have FPs, thereby affirming the utility of modifications to the Wardowski contraction concept. Our research extends the work of Nadler [26] by integrating real-world examples and applications. Compared to previous results in $b$-MS, our findings offer a broader scope of applicability.

## Acknowledgments

The authors are grateful to reviewers for their useful remarks which helped us to improve the presentation of this manuscript.

## Author Contributions

**Conceptualization:** Hamed Al Sulami.

**Formal analysis:** Hamed Al Sulami.

**Supervision:** Afshan Batool, Aftab Hussain.

**Writing – original draft:** Maryam Iqbal.

**Writing – review & editing:** Aftab Hussain.

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
