## [Decision Letter · Decision Letter 0]

30 Aug 2024

PONE-D-24-31370Generalized Wardowski Type Contractive Mappings in b-Metric Spaces and Some Fixed Point Results with Applications in Optimization Problem and Modeling Biological EcosystemPLOS ONE

Dear Dr. Aftab,

Thank you for submitting your manuscript to PLOS ONE. After careful consideration, we feel that it has merit but does not fully meet PLOS ONE’s publication criteria as it currently stands. Therefore, we invite you to submit a revised version of the manuscript that addresses the points raised during the review process.

We look forward to receiving your revised manuscript.

Kind regards,

Naeem Saleem

Academic Editor

PLOS ONE

Journal Requirements:

Reviewers' comments:

Reviewer's Responses to Questions

**Comments to the Author**

1. Is the manuscript technically sound, and do the data support the conclusions?

Reviewer #1: No

Reviewer #2: Yes

Reviewer #3: Yes

Reviewer #4: Partly

2. Has the statistical analysis been performed appropriately and rigorously? 

Reviewer #1: No

Reviewer #2: N/A

Reviewer #3: Yes

Reviewer #4: N/A

3. Have the authors made all data underlying the findings in their manuscript fully available?

Reviewer #1: No

Reviewer #2: Yes

Reviewer #3: Yes

Reviewer #4: No

4. Is the manuscript presented in an intelligible fashion and written in standard English?

Reviewer #1: No

Reviewer #2: Yes

Reviewer #3: Yes

Reviewer #4: No

5. Review Comments to the Author

Reviewer #1: The paper is written badly as it is very difficult to read due to non-standard symbols. There is no novelty in the paper as more generalized spaces are available in the literature. Authors has claimed for application but there is no details studies have been done. The paper has NO merit to publish in any SCI/SCOPUS journals.

Paper is Rejected from bottom.

Reviewer #2: The manuscript is well written and the authors have applied the concept of Wardowski's F-contraction to prove some fixed point results in b-metric spaces and applied them in modeling biological ecosystem. The paper may be accepted after the following revision.

1. In the introduction, add some discussion on the importance of studying b-metric spaces and its difference with metric spaces.

2. The notations may be simplified.

3. Check out thoroughly for some undefined symbols.

4. Theorem 3.1 should be moved to introduction.

5. In the abstract, emphasize the original contribution of this paper.

Reviewer #3: Generalized Wardowski Type Contractive Mappings in b-Metric Spaces and Some Fixed Point Results with Applications in Optimization Problem and Modeling BiologicalEcosystem

Key Contributions

1. The paper proposes a β-(θ, ϑ) a generalizedWardowski type quasi-contractionto split fixed point problems.

2. First, authors use our new contraction to provide typical fixed point results, and then we show that there is a generalized quasi-contraction of the Wardowski type, confirming the validity of our results.

3. Second they demonstrate the modelling of biological ecosystems using Nadler’s work and the application of our results to an optimization problem.

4. In order to demonstrate the effectiveness of results, and present a comparison between result and Nadler’s work.

5. Theorem has important applications in computer science, biology, and economics, among other domains. Biological ecosystem modelling.

Areas of Application

• Computer vision

• Computer graphics

• Image restoration

• Complex 3D Models

Comments and suggestions:

Abstract:

Generalised should be generalized, optimisation should be optimization,

Introduction: “Banach withdrawal standard” revise this sentence

Line -4: Aftab et al should be Hussain et al.

Section 3.2 heading or in contents “optimisation should be optimization”

This paper is a solid contribution to the field of nonlinear analysis, particularly in the context of fixed point problems.

Conclusion: A minor revision is required

Reviewer #4: Generalized Wardowski Type Contractive Mappings in b-Metric Spaces and Some Fixed Point Results with Applications in Optimization Problem and Modeling Biological Ecosystem

Authors: Maryam Iqbal , Afshan Batool , Aftab Hussain , Hamed Al-Sulami

The paper claims to introduces a generalized Wardowski-type quasi-contraction, termed \\beta-(\\theta,\\ \\vartheta), within the context of b-metric spaces. The work establishes fixed point results using this new contraction and confirms its validity by identifying it as a generalization of Wardowski's quasi-contraction. The study further claims to illustrate the application of these results to biological ecosystem modeling, based on Nadler's work, and to an optimization problem. While the mathematical aspects of the paper may have some merit, the claimed applications are speculative rather than genuine. To present real applications, the authors should develop concrete models and revise these sections with a more constructive application(s) or remove the application parts altogether. The paper is also poorly organized and written in weak English. Therefore, I recommend major revision, and here are my further observations:

In the introduction, the authors used the phrase `which is referred as’ this should be `which is referred to as’, please correct this throughout the manuscript.

Add more related literature in the introduction and discuss in depth about each work.

I was expecting to see the organization or structure of the paper at the end of the introductory section but there is none. Please, include this.

The first sentence in the Preliminaries section is grammatically not correct, please correct it.

Definition 2.1 is also poorly written.

In the entire paper the name ‘Nadler’ was written wrongly almost everywhere, see, for example second line under main results and the subsection 3.1.

Optimization was also written as optimisation in some parts of the paper, please be specific and rewrite them correctly, see, for example the title of the paper and subsection 3.2.

I could not find how in subsection 3.1, a biological ecosystem was modeled using Nadler’s work, please put a genuine application or remove this claim.

Subsection 3.2 is not an application rather than just speculations, please remove it.

The paper should be thoroughly reorganized, proofread, and spell-checked.

6. PLOS authors have the option to publish the peer review history of their article (what does this mean?). If published, this will include your full peer review and any attached files.

Reviewer #1: No

Reviewer #2: **Yes: **Pradip Debnath

Reviewer #3: No

Reviewer #4: No

---

## [Author Response · Author response to Decision Letter 0]

29 Sep 2024

We have incorporated the point-by-point comments and suggestions of the reviewers, and for more details see the response letter.

---

## [Decision Letter · Decision Letter 1]

17 Oct 2024

Generalized Wardowski Type Contractive Mappings in b-Metric Spaces and Some Fixed Point Results with Applications in Optimization Problem and Modeling Biological Ecosystem

PONE-D-24-31370R1

Dear Dr. Hussain,

We’re pleased to inform you that your manuscript has been judged scientifically suitable for publication and will be formally accepted for publication once it meets all outstanding technical requirements.

Kind regards,

Naeem Saleem

Academic Editor

PLOS ONE

Reviewers' comments:

Reviewer's Responses to Questions

**Comments to the Author**

1. If the authors have adequately addressed your comments raised in a previous round of review and you feel that this manuscript is now acceptable for publication, you may indicate that here to bypass the “Comments to the Author” section, enter your conflict of interest statement in the “Confidential to Editor” section, and submit your "Accept" recommendation.

Reviewer #2: All comments have been addressed

Reviewer #4: All comments have been addressed

2. Is the manuscript technically sound, and do the data support the conclusions?

Reviewer #2: Partly

Reviewer #4: Yes

3. Has the statistical analysis been performed appropriately and rigorously? 

Reviewer #2: N/A

Reviewer #4: Yes

4. Have the authors made all data underlying the findings in their manuscript fully available?

Reviewer #2: Yes

Reviewer #4: Yes

5. Is the manuscript presented in an intelligible fashion and written in standard English?

Reviewer #2: Yes

Reviewer #4: Yes

6. Review Comments to the Author

Reviewer #2: The revised version is much improved. May be accepted. The authors have addressed the queries well in the revised version.

Reviewer #4: The revised manuscript is satisfactory and has addressed all the issues by the reviewers, so the paper can be accepted.

7. PLOS authors have the option to publish the peer review history of their article (what does this mean?). If published, this will include your full peer review and any attached files.

Reviewer #2: **Yes: **Pradip Debnath

Reviewer #4: No

---

## [Editor Report · Acceptance letter]

5 Dec 2024

PONE-D-24-31370R1 

PLOS ONE

Dear Dr. Hussain, 

I'm pleased to inform you that your manuscript has been deemed suitable for publication in PLOS ONE. Congratulations! Your manuscript is now being handed over to our production team.

Kind regards, 

on behalf of

Professor Dr. Naeem Saleem 

Academic Editor

PLOS ONE